# Rejuvenation of Zr-Based Bulk Metallic Glasses by Ultrasonic Vibration-Assisted Elastic Deformation

**DOI:** 10.3390/ma13194397

**Published:** 2020-10-02

**Authors:** Yan Lou, Shenpeng Xv, Zhiyuan Liu, Jiang Ma

**Affiliations:** Guangdong Provincial Key Laboratory of Micro/Nano Optomechatronics Engineering, College of Mechatronics and Control Engineering, Shenzhen University, Shenzhen 518060, China; 1810293008@email.szu.edu.cn (S.X.); zyliu@szu.edu.cn (Z.L.); majiang@szu.edu.cn (J.M.)

**Keywords:** bulk metallic glasses (BMGs), rejuvenation, ultrasonic vibration-assisted elastic deformation (UVEF), free volume, plasticity

## Abstract

The rejuvenation of Zr_52.5_Cu_17.9_Ni_14.6_Al_10_Ti_5_ bulk metallic glasses (BMGs) by ultrasonic vibration-assisted elastic deformation (UVEF) was studied herein. The UVEF-treated samples demonstrate an obvious rejuvenation and have a higher relaxation enthalpy and a smaller range of supercooled liquid regions than the as-cast samples. The fracture of the rejuvenated amorphous alloy is mainly ductile fracture, and shear deformation occurs in the deformation region. It is also found that as the amplitude increases, the free volume of the rejuvenated amorphous alloy increases, the yield strength and the elastic modulus decrease, and the formability increases. The free-volume content is used to characterize the degree of rejuvenation, and a mathematical model of the relationship between the ultrasonic amplitude and free volume is established. In addition, it is found that the ultrasonic vibration stress induces the additional free volume in the Zr-based bulk metallic glasses and improves the plasticizing behavior. The temperature rise caused by the ultrasonic thermal effect does not induce additional free volume.

## 1. Introduction

Bulk metallic glasses (BMGs) are materials with excellent performance and mechanical properties, such as high yield strength [1], high toughness [2], high hardness [3] and exceptional “damage tolerance” [4,5]. However, the nonuniform deformation caused by the highly localized deformation at room temperature directly causes its brittleness [6,7], which has become the bottleneck for the widespread application of BMGs. Reducing the brittleness of BMGs has become a scientific issue that scholars have begun to pay attention to.

The rejuvenation of amorphous alloys occurs when BMGs are injected with high energy because it endows them with additional free volume and greater plasticity [8,9]. Therefore, rejuvenation of BMGs to reduce their brittleness and improve their plasticity at room temperature has attracted a lot of attention [10,11,12]. For example, J. Michler [13] conducted compression experiments on ion-radiated BMGs and found that they can increase the free-volume content of BMGs and enhance their plasticity. J. Das [14] utilized cold-rolled Zr-based BMGs at room temperature to overcome their inherent brittleness and make BMGs plastic from 0.5% to 15%. F. X. Li [15] discovered that during the elastic compression of Zr_35_Ti_30_Be_27.5_Cu_7.5_ BMGs at room temperature, the BMGs transitioned from mechanical relaxation to rejuvenation, thereby improving their plasticity. W. Guo [16] studied the rejuvenation of Zr–Cu–Al–Ni–Ta BMGs under cryogenic cycle treatment (DCT) and found that by adding more Ta, the BMGs can be restored to an elevated energy, and the compressive fracture strength and plastic strain increased with increasing Ta content. Through plastic deformation under triaxial compression at room temperature, Y. Li and A.L. Greer [4] rejuvenated BMGs samples and found that the rejuvenated amorphous material showed excellent plastic deformation ability. However, elastic loading [17], ion radiation [18], thermal cycling [19,20], and plastic deformation [21] methods require a substantial amount of time to rejuvenate the BMGs, and during this time, a relaxation effect inevitably occurs, weakening the rejuvenation effect.

In addition, ultrasonic vibration-assisted has an ultrasonic stress effect and ultrasonic thermal effect. P. Chen [22] used ultrasonic vibration to process a Ti-based amorphous powder to make nanocrystalline bulk materials. The amorphous powder was crystallized by the frictional heat generated during the ultrasonic vibration and then welded to form a bulk nanocrystalline material. J. Ma et al. [23] used supersonic vibration to stamp BMGs and found that this forming method completed thermoplastic forming in seconds and largely avoided the time-dependent crystallization and oxidation processes, thus avoiding the risk of crystallization that occurs during traditional heat processing. N. Li et al. [24] conducted uniaxial tensile and compression experiments on Zr-based amorphous alloys under the action of a vibration field and found that with increasing vibration frequency, the free-volume content of the amorphous alloy increased and the flow unit volume decreased, which caused the flow viscosity to decrease and the micro-forming ability to increase. However, there are no reports that specifically analyze the influence of ultrasonic vibration stresses and ultrasonic thermal effects on amorphous properties.

A novel method of ultrasonic vibration-assisted elastic deformation (UVEF) was developed that can successfully and rapidly rejuvenate Zr-based BMGs within 8 s. It was found that ultrasound-assisted vibration can rapidly increase the internal energy of BMGs and quickly drive the loosely packed atoms in BMGs to high-energy regions, thereby driving additional areas with free volume and rheological units to form shear bands. These processes result in the rapid rejuvenation of BMGs and prevent the time-dependent crystallization and relaxation phenomena. In addition, the thermodynamics, mechanical properties and fracture morphology of the rejuvenated UVEF-treated samples were analyzed. It was found that as the amplitude increased, the yield strength and elastic modulus of the UVEF-treated samples decreased, and the plasticity was greatly improved. The free volume was used to characterize the degree of rejuvenation of the amorphous alloys herein, and a mathematical model of the relationship between the ultrasonic amplitude and the free volume of the rejuvenated BMGs was established. The effects of the temperature rise and stress caused by the ultrasonic vibration on the rejuvenation properties of BMGs were analyzed. These findings provide a convenient and fast method to reduce the room-temperature brittleness of BMGs and improve their plasticity.

## 2. Experimental Methods

### 2.1. Sample Preparation

The experimental material is an amorphous alloy with an alloy composition of Zr_52.5_Cu_17.9_Ni_14.6_Al_10_Ti_5_ (atomic percentage). The raw materials of metals Zr, Cu, Ni, Al, and Ti with a purity of 99.9% or more are prepared according to the nominal composition. An electric arc furnace is used to melt the master alloy, and then a copper die suction casting method is used to prepare rod-shaped amorphous particles with a diameter of Φ 2 mm. A low-speed diamond cutting machine is used to process a cylindrical sample with dimensions of Φ 2 mm × 4 mm. The ends of the cylinder are also polished to ensure that both ends are parallel to each other and orthogonal to the centerline of the cylinder. The sample was analyzed with X-ray diffraction (XRD, Bruker D8 Advance, Bruker, Karlsruhe, Germany) to ensure that the sample is in an amorphous state.

### 2.2. UVEF Processing

The experiments herein use a homemade ultrasonic vibration-assisted compression test platform. The ultrasonic vibration-assisted device is installed on a Zwick Z050 tensile testing machine (Zwick Roell Group, Ulm, Germany). The maximum power of the ultrasonic system is 1500 W, and the vibration frequency is 20 kHz. Due to the limitation of the experimental conditions, the ultrasonic frequency in this study is unchanged; only the amplitude is changed, and the amplitudes are 19, 27, 36, and 43 μm. The experiment is divided into two steps. First, the sample is positioned and clamped by a fixture. Next, an ultrasonic vibration is applied to the sample while the fixture and the sample are fixed (Figure 1). The speed is V = V_0_ + V_ul_ (t), where V_0_ is the pressing speed, V_ul_ is the ultrasonic vibration speed, and t is the time. A strain rate of $\dot{\varepsilon }\;=$ 0.01 s^−1^ and a strain of ε = 2% were used to ensure compression within the amorphous elastic deformation range. In addition, during the ultrasonic vibration-assisted elastic compression deformation, a high-frequency vibration causes the samples to slip, so a preload force of 50 N is applied first to fix the samples, and then the ultrasonic vibration is applied until the elastic compression deformation ends, as shown in Figure 1. The time required for UVEF can be calculated:(1)$$t=t_{pl}+t_{ul}=\frac{S_{pl}}{V_{tp}}+\frac{\varepsilon }{\dot{\varepsilon }}$$
where $t_{pl}$ is the time of preloading, $t_{ul}$ is the ultrasonic loading time, $S_{pl}$ is the distance between the initial punch and the upper surface of the sample, and $V_{tp}$ is the preloading speed. $S_{pl}$ and $V_{tp}$ were taken as 0.5 mm and 0.1 mm/s, respectively. A strain rate of $\dot{\varepsilon }\;=$ 0.01 s^−1^ and a strain of ε = 2% were applied to ensure that compression is within the amorphous elastic deformation range, and BMGs can be rejuvenated within 8 s. Five BMGs samples were repeatedly compressed under each deformation process condition.

### 2.3. Hot-Compressed Elastic Deformation (HEF) Experiment

To specialized analyze the effects of the temperature rise and stress produced during the ultrasonic vibration on the rejuvenation properties of the amorphous alloys, we eliminate the effect of ultrasonic stress, and only the effect of temperature rise is studied separately. The temperature rise at different amplitudes is obtained according to previous research results [25]. That is, amplitudes of 19, 27, 36, and 43 μm correspond to temperature increases of 80, 150, 200, and 270 °C. HEF experiments at the corresponding temperatures are performed on the thermal simulation machine (Gleeble3800, DSI, Austin, TX, USA). The corresponding strain rate and strain are consistent with the UVEF experiment and are 0.01 s^−1^ and 2%, respectively. Five BMGs samples are repeatedly hot-compressed under each deformation condition.

### 2.4. Analytical Testing

To study the properties of the Zr-based BMGs samples, a series of analytical tests are performed on them. XRD (Bruker D8 Advance, Bruker, Karlsruhe, Germany) is used to determine the crystallization of the samples. The scanning range is 20°~80°, the step size is 0.02°∙step^−1^, and the scanning speed is 12°∙min^−1^. The Vickers microhardness test (BUEHLER 5103, Buehler, Lake Bluff, IL, USA; ASTM E92-82, West Conshohocken, PA, USA) is used to determine the hardness of the samples, the diamond indenter was pressed into the surface of the specimen along its axial direction with a force of 50 gf, and then the force was held for 10 s. Differential scanning calorimetry (DSC, PerkinElmer DSC8000, PerkinElmer, Poulsbo, WA, USA) is used to analyze the thermodynamic characteristics of the samples at a heating temperature of 520 °C, and the heating and cooling rates are 20 °C min^−1^. Transmission electron microscopy (TEM, JEM-2100F, JEOL Ltd., Tokyo, Japan) with an acceleration voltage of 200 kV is used to observe the microstructure of the samples. The thin foil samples used for TEM observation are cut from the samples and mechanically ground to a thickness of 0.5 mm, then the sample was mounted on a TEM copper grid. The thinning procedures were performed by alternately milling both sides of the lamina with a tilt angle of ±1.2°. The thickness of lamina was reduced within 100 nm with a (Ga) ion energy of 30 kV and a current of 0.1 nA. The quasi-static compression fracture test is used to determine the plasticity, yield strength and elastic modulus of the samples. The experiments are carried out on a Zwick Z050 universal tensile machine (Zwick Roell Group), and the strain rate is 0.0005 s^−1^. Scanning electron microscopy (SEM, FEI Quanta 450FEG, FEI Company, Hillsboro, OR, USA) is used to observe the fracture morphology of the quasi-static compression fracture samples. All test experiments are repeated 5 times and the average is reported herein.

## 3. Results and Discussion

Figure 2 is a comparison of the XRD patterns of the as-cast and UVEF-treated samples. It can be found that the XRD diffraction peaks of the UVEF-treated samples are consistent with those of the as-cast samples. There are broad diffusion peaks without any obvious crystalline peaks in both cases, which indicates that the as-cast and UVEF-treated samples are all amorphous, and ultrasonic vibration-assisted elastic deformation does not cause the amorphous material to crystallize. Regarding the naming convention, the UVEF-19 sample is an amorphous sample treated by ultrasonic vibration-assisted elastic deformation with an amplitude of 19 μm.

To further confirm the amorphous properties of the as-cast and UVEF-treated samples, high-resolution TEM is used to obtain an image of the microstructure of the sample, which is shown in Figure 3. No nanocrystals are found in the as-cast samples or the UVEF-treated samples, and the microstructure of all samples shows labyrinth-like characteristics similar to those of an amorphous state, which is consistent with the XRD test results.

### 3.1. Thermodynamics Analysis

The as-cast and UVEF-treated samples are analyzed by DSC, and the thermodynamic properties of the samples are obtained from the DSC curve. As shown in Figure 4, the DSC curves of all samples show an endothermic phenomenon starting from the T_g_ temperature, which is a characteristic of glass transition, and then exhibit an exothermic phenomenon corresponding to the crystallization behavior to the T_x_ temperature. The values of T_g_ and T_x_ for as-cast and UVEF treated samples as shown in Table 1.

Compared with that of the as-cast sample, the crystallization temperature T_x_ of the UVEF-treated sample shows only slight fluctuations, roughly in the range of 453 ± 1 °C, while the glass transition temperature T_g_ showed an upward trend with increasing ultrasonic amplitude. This shows that an increased amplitude ultrasonic vibration can increase the glass transition temperature T_g_ of the amorphous alloy and reduce the range of its supercooled liquid region. According to the Spaepen plastic deformation mechanism, BMGs will undergo uneven plastic deformation at high strain rates, resulting in a large number of shear bands. The arrangement density of atoms near the shear zone is less than that of the matrix, so a large amount of free volume is concentrated in the shear zone at this time, causing the increase of atomic mobility, the deterioration of the thermal stability of the material, and the increase of T_g_ [8,26].

Because of the structural relaxation, the amorphous alloy exhibits exothermic phenomena during continuous heating to the T_g_ (Figure 5). The larger the value of free volume change (ΔV_f_) is, the greater the initial free volume (V_f_) of the amorphous alloy before heating [27]. Figure 5 shows the relaxation enthalpy ΔH of the as-cast and UVEF-treated samples during DSC thermodynamic analysis. The ΔH is calculated by calculating the integrated area of the exothermic peak, and the change in ΔH reflects the disorder of the atomic arrangement [28]. It is found that with an increase in the ultrasonic amplitude, the ΔH value of the UVEF-treated sample increases significantly. The larger the ΔH value is, the greater the free volume V_f_ value of the amorphous alloy. This is because under the vibration load, the UVEF-treated samples were injected with a high energy and the atom diffusion increases. The atomic arrangement also becomes looser, and the larger the amplitude is, the stronger the atom diffusion and the higher the amorphous energy. This result also shows that the degree of rejuvenation of the UVEF-treated sample increases.

### 3.2. Macromechanics Analysis

The compressive fracture stress–strain curves of the as-cast and UVEF-treated samples are shown in Figure 6a. The as-cast sample exhibits typical brittle failure characteristics. There is no obvious yield stage in the stress–strain curve prior to fracture, and the sample fails immediately after reaching the maximum stress (approximately 1800 MPa). After applying the ultrasonic elastic deformation treatment, the failure mode of the UVEF-treated sample changes, and the yield characteristics appear after the elastic deformation stage. That is, a plastic strain was generated, and the maximum strain is 14%, which does not include the elastic strain. With the increase of ultrasonic amplitude, the yield stress (σ_s_) decreased from 1736 to 1455 MPa, the elastic modulus (E) also decreased to about 40 GPa (Figure 6b), and the elongation (Δ) increased to 12% (Figure 6c).

### 3.3. Micromechanics Analysis.

Generally, the hardness of amorphous alloys is closely related to the amount of free volume [29]. Figure 7 shows the microhardness values of the as-cast and UVEF-treated samples. The microhardness of the UVEF-treated sample is significantly lower than that of the as-cast sample. The decrease in microhardness is due to the increase in the atomic spacing of the amorphous alloy. The combined energy of the metal bonds is weakened and the amount of free volume increases [30], which also shows that ultrasonic vibration-assisted elastic deformation can increase the amount of free volume and restore desirable properties. At the same time, it is found that as the amplitude increases, the microhardness of the UVEF-treated samples decreases slightly.

### 3.4. Fracture Microtopography

Figure 8 shows the compressive fracture micromorphology of the as-cast and UVEF-treated samples obtained by SEM. All samples fracture along the maximum shear stress plane at an angle of approximately 45° to the loading axis. The microstructure was not uniform across the fracture surface, and the fractures of as-cast samples are obviously in the shape of veins, mountains, and rivers (Figure 8a), which is a typical brittle fracture morphology. The fracture surface morphology of the UVEF-treated samples indicates mainly ductile fracture (Figure 8b–e). The fracture surface has a large number of vein-like textures. This texture is due to the shear zone breaking during the stress loading process. The local melting phenomenon caused by the instantaneous release indicates that the material has undergone shear deformation in the region, which is consistent with the experimental results in Figure 6a, indicating that the ultrasonic vibration-assisted elastic deformation process rejuvenates the amorphous alloys.

### 3.5. Properties of the Hot-Compressed Elastic Deformation Treated Sample

From a comparison of XRD patterns of hot-compressed elastic deformation (HEF)-treated samples in Figure 9a, it can be found that the HEF-treated samples have broad diffusion peaks and no obvious crystalline peaks, which proves that the HEF-treated samples are still all amorphous. Regarding the sample naming convention, HEF-80 refers to amorphous samples processed by hot-compressed elastic deformation at a temperature of 80 °C. The microscopic morphology observed with TEM shown in Figure 9b indicates that the HEF-treated sample shows a maze-like amorphous structure, and no nanocrystals are found.

Figure 10a shows the microhardness values of the as-cast and HEF-treated samples. The microhardness value does not decrease with increasing temperature. Compared with the UVEF-treated samples (Figure 7), the effect of deformation on the amorphous particles is the opposite, indicating that only an elastic deformation treatment with increasing temperature does not induce an increase in the amount of free volume. It is also found that the relaxation enthalpy ΔH of the HEF-treated sample is slightly larger than that of the as-cast sample (Figure 10b), but as the temperature increases, the relaxation enthalpy ΔH of the HEF-treated sample essentially does not change, indicating that its free volume does not change with increasing temperature. It also further shows that the aspect of the ultrasonic vibration-assisted elastic deformation process that creates additional free volume is the ultrasonic vibration stress, not the temperature rise caused by the ultrasonic thermal effect. The ultrasonic vibration stress can create additional free volume and improve the rejuvenation degree of amorphous alloys [31].

It is found that the fracture stress–strain curve of the HEF-treated samples has a very short yield stage before fracture, approximately 0.3% of the strain, and then the samples immediately fracture (Figure 11). At the same time, it is found that the yield strength, elastic modulus, and plastic strain of the HEF-treated sample essentially did not change with increasing temperature. This also verifies the results of Figure 10. The increase in the plasticity of the UVEF-treated sample is caused by an increase in the free volume induced by the ultrasonic vibration stress, which promotes the rejuvenation of the amorphous alloy. The ultrasonic thermal effect that causes a temperature increase does not increase the plasticity of the amorphous sample. This is because the maximum temperature rise caused by the ultrasonic thermal effect in this study is 270 °C, which is far lower than the T_g_ of the Zr-based BMGs. Therefore, at low temperatures, an increase in the temperature cannot increase the free volume. The ultrasonic vibration stress is the main reason for the increase in the free volume and the degree of rejuvenation demonstrated herein [31].

### 3.6. Mathematical Model of the Relationship between Ultrasonic Amplitude And Free Volume

The shear phase transformation zone (STZ) model [32] is widely used to describe the physical process of plastic deformation of amorphous alloy materials. This STZ model is applicable whether the deformation is transient or time-dependent and uniform or nonuniform. Generally, shear deformation is preferentially activated in areas with an increased free volume. Therefore, an increased free volume leads to easier and more shear deformation, which in turn leads to better plasticity.

It is deduced from this that the BMGs with an increased free volume show a decreased resistance to any form of deformation. The ultrasonic amplitude leads to a high-energy state, which increases the free volume and increases the degree of rejuvenation of amorphous alloys.

To quantify the effect of the ultrasonic amplitude on the rejuvenation degree of the amorphous samples, the amount of free volume is used to characterize the rejuvenation degree of the amorphous alloys, so the amount of free volume in the UVEF-treated samples is theoretically calculated based on the free volume formula [33]:(2)$$v_{f}=v_{f}^{*}-\frac{v_{f}^{*}}{\overset{\cdot }{\gamma }}\exp(-\frac{2s+c}{v_{f}^{*}})$$
where *C* is the fitting parameter, $v_{f}^{*}$ is the steady-state free volume,$\overset{\cdot }{\gamma }$ is the shear strain rate, and *s* is the strain. Also:(3)$$v_{f}^{*}={\left[\frac{1}{v_{fe}}-(k_{t}\ln\overset{\cdot }{\gamma }+l)\right]}^{-1}$$
(4)$$v_{fe}=\frac{T-T_{0}}{DT_{0}}$$
where *v_fe_* is the equilibrium free volume; *k_t_* = 26.6 + 0.044*T* and *l* = 43.9 + 0.082*T* represent temperature-related parameters [33]; *D* is the brittleness index of the BMGs, which is taken as 18.5; and *T*_0_ is the Vogel–Fulcher temperature, *T*_0_ = 2/3*T_g_*.

The amount of free volume of the UVEF-treated sample can be obtained according to Equation (2), as shown in Figure 12. It can be found that the larger the amplitude of the UVEF-treated sample is, the greater the free volume. Finally, using polynomial fitting, the coefficient of determination for curve fitting is 0.99983, the mathematical model to obtain the ultrasonic amplitude (*A*) and free volume (*v_f_*) is obtained as follows:(5)$$v_{f}=7.9857-0.20477A+0.01216A^{2}$$

As the amplitude increases, the amount of free volume increases sharply, indicating that the greater the ultrasonic energy is, the greater the degree of amorphous rejuvenation.

## 4. Conclusions

Compared with metallic glass rejuvenation methods such as elastic loading and ion radiation, UVEF has a short processing time (8s), will not cause damage, and is also controllable. It is found that the relaxation enthalpy increases and the range of the supercooled liquid region decreases when Zr based metallic glasses are UVEF-treated at room temperature. With the increase of amplitude, the free volume and the formability of the UVEF-treated sample increase, and the yield strength and elastic modulus decrease. Ultrasonic vibration stress is the main reason for the increase in the free volume of Zr-based BMGs.

## Figures and Tables

**Figure 1 materials-13-04397-f001:**
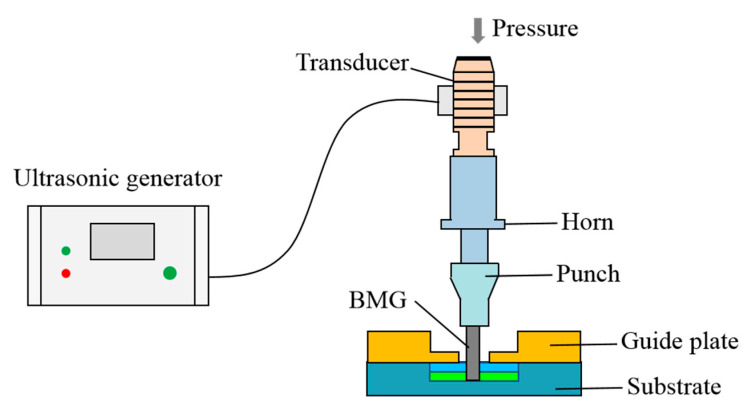
Ultrasonic vibration-assisted elastic deformation diagram.

**Figure 2 materials-13-04397-f002:**
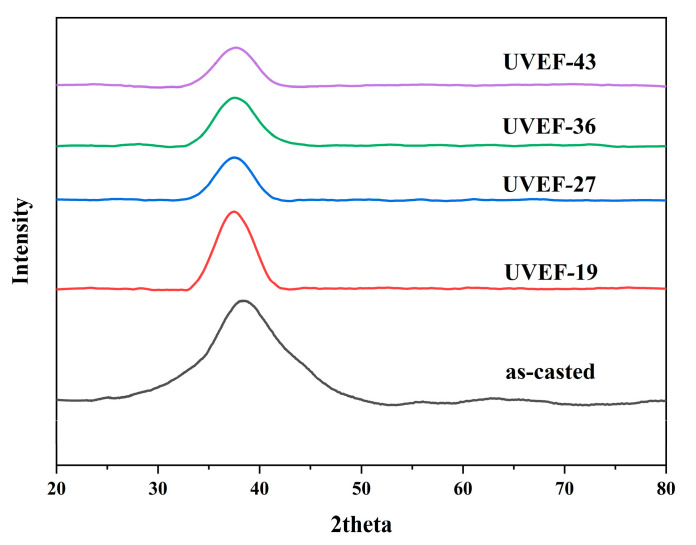
Comparison of the X-ray diffraction (XRD) patterns of the as-cast and ultrasonic vibration-assisted elastic deformation (UVEF)-treated samples.

**Figure 3 materials-13-04397-f003:**
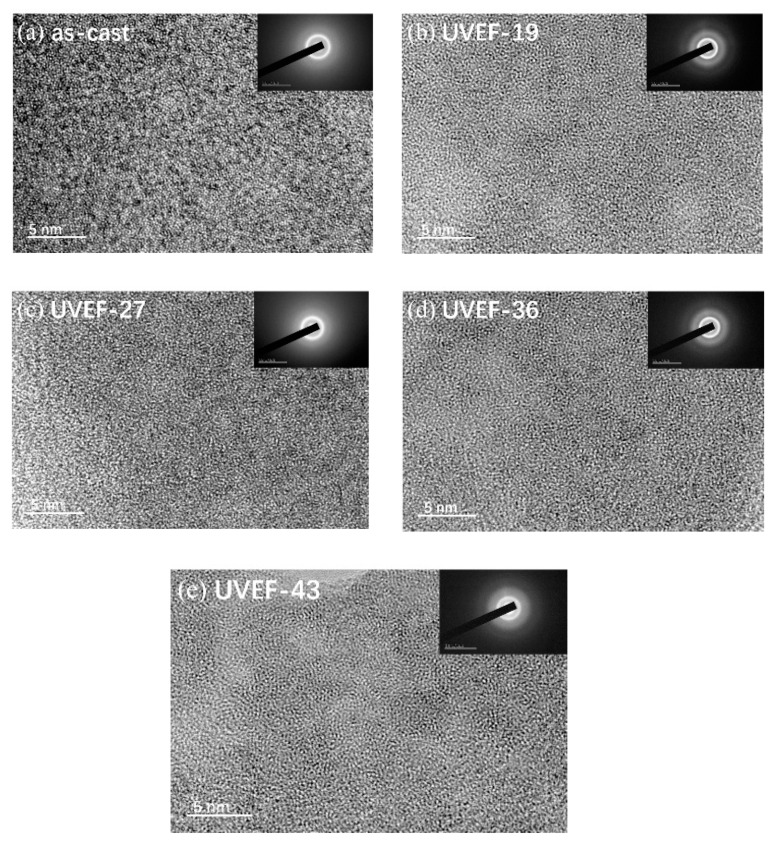
Transmission electron microscopy (TEM)) images of the as-cast sample and the UVEF-treated samples: (**a**) as-cast; (**b**) UVEF-19; (**c**) UVEF-27; (**d**) UVEF-36; and (**e**) UVEF-43.

**Figure 4 materials-13-04397-f004:**
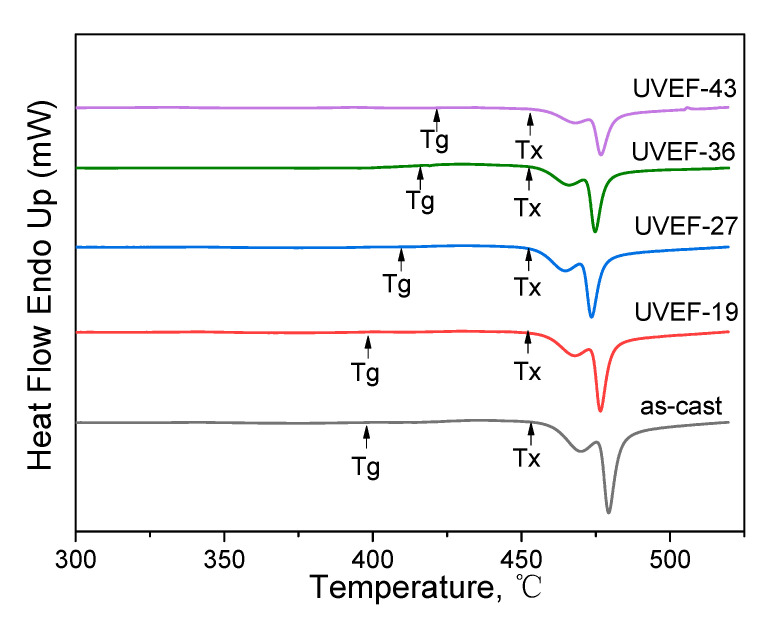
Differential scanning calorimetry (DSC) curves of the as-cast and UVEF-treated samples.

**Figure 5 materials-13-04397-f005:**
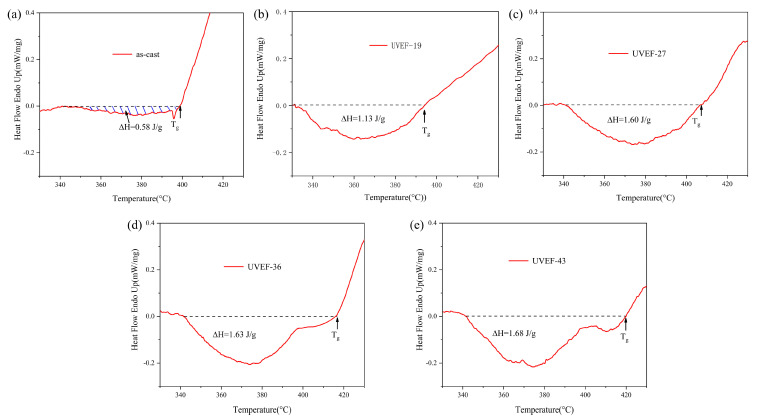
Relaxation enthalpy ΔH of the as-cast and UVEF-treated samples (**a**) as-cast; (**b**) UVEF-19; (**c**) UVEF-27; (**d**) UVEF-36; and (**e**) UVEF-43.

**Figure 6 materials-13-04397-f006:**
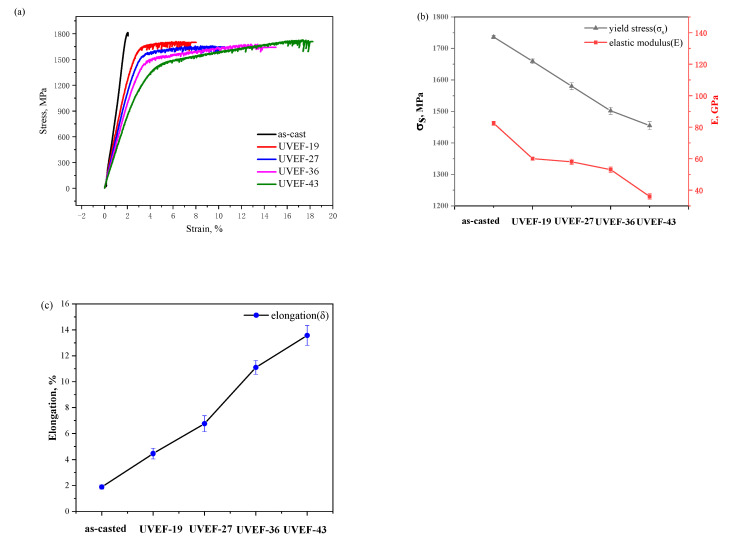
(**a**) Compression fracture stress–strain curves of as-cast and UVEF-treated samples (**b**) values of the yield stress and elastic modulus, and (**c**) elongation of the as-cast and UVEF treated samples.

**Figure 7 materials-13-04397-f007:**
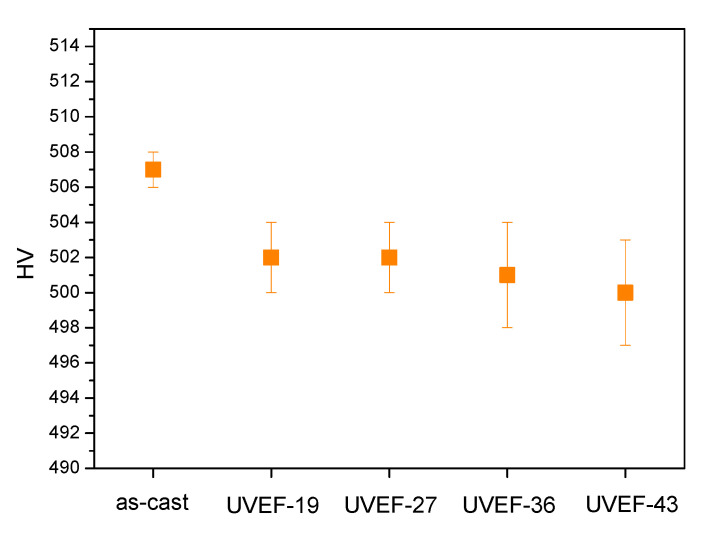
Comparison of the microhardness of the as-cast and UVEF-treated samples.

**Figure 8 materials-13-04397-f008:**
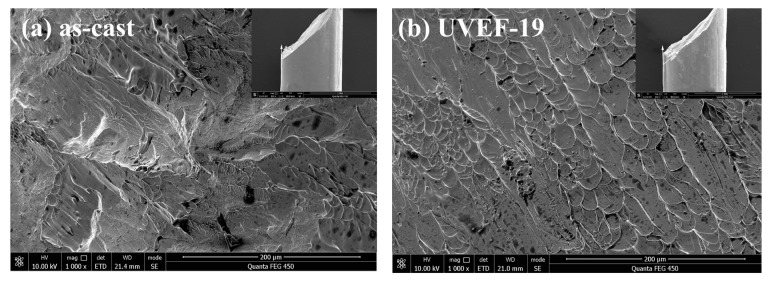
Compressive fracture micromorphology of the as-cast and UVEF-treated samples: (**a**) as-cast; (**b**) UVEF-19; (**c**) UVEF-27; (**d**) UVEF-36; and (**e**) UVEF-43.

**Figure 9 materials-13-04397-f009:**
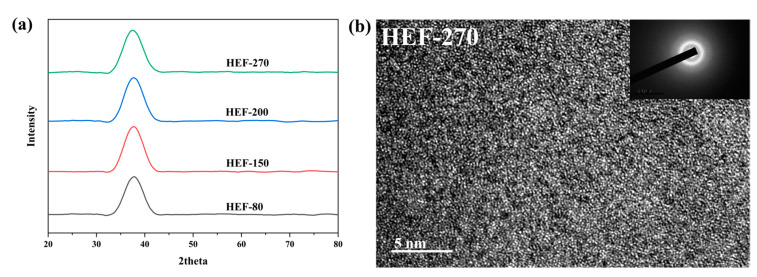
(**a**) Comparison of XRD patterns of as-cast and hot-compressed elastic deformation (HEF) -treated samples, and (**b**) TEM image of HEF-270 treated sample.

**Figure 10 materials-13-04397-f010:**
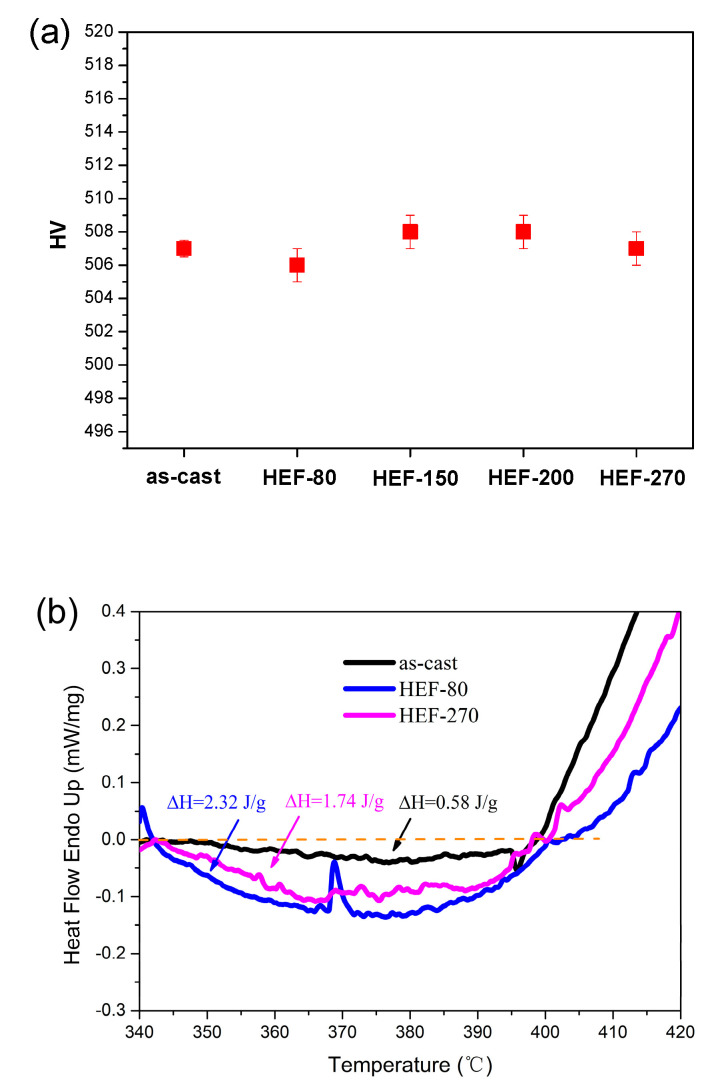
(**a**) Comparison of microhardness of the as-cast and HEF-treated samples and (**b**) comparison of relaxation enthalpies of the as-cast and HEF-treated samples.

**Figure 11 materials-13-04397-f011:**
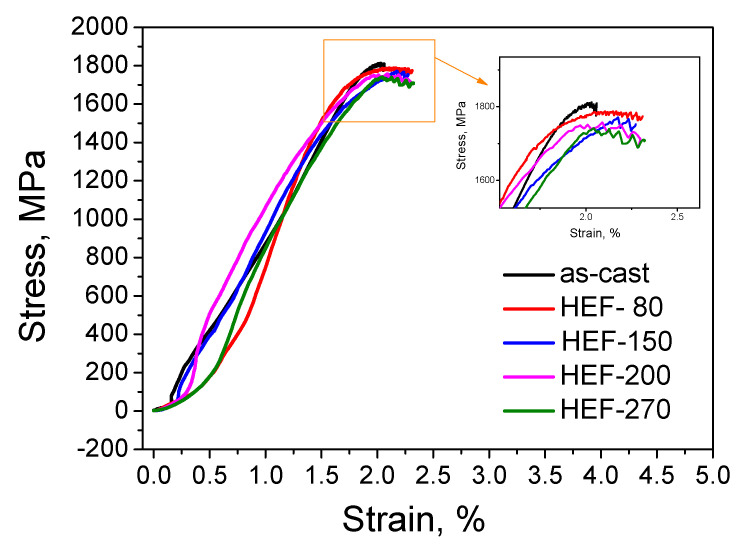
Fracture stress–strain curve of the as-cast and HEF-treated samples.

**Figure 12 materials-13-04397-f012:**
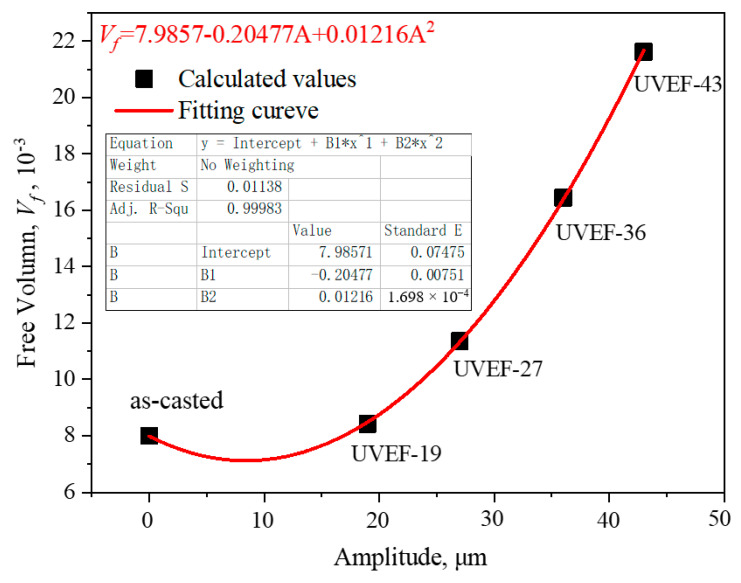
Theoretical model of the relationship between the free volume and amplitude of UVEF-treated samples.

**Table 1 materials-13-04397-t001:** The values of the glass transition temperature T_g_ and the crystallization temperature T_x_ for as-cast and UVEF treated samples.

Sample	T_g_ (K)	T_x_ (K)	ΔT (K)
As-cast	397	452	55
UVEF-19	398	452	54
UVEF-27	410	453	43
UVEF-36	416	453	37
UVEF-43	422	454	32

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
