# Peer review of "Rejuvenation of Zr-Based Bulk Metallic Glasses by Ultrasonic Vibration-Assisted Elastic Deformation"

_materials, 2020, doi:10.3390/ma13194397_

Round 1

Reviewer 1 Report

In this work authors systematically studied the rejuvenation effect of ultrasonic vibration on Zr52.5Cu17.9Ni14.6Al10Ti5 bulk metallic glass. The Authors used XRD, DSC, and TEM experiments to characterize the samples. The results were analyzed systematically and also discussed in depth. These results are interesting enough to publish in Materials, however, there are some minor issues needed to be addressed before the official acceptance.

Minor revision Recommended

Comments to Author

  • Inline 62, the authors mentioned the UVEF method can be used to rejuvenate the Zr-based BMG samples within 8 s. however in experiment nowhere the parameter time is considered or discussed. So how did the authors come to this conclusion.?
  • Figure 1. Is not clear, from which side, (top or bottom) they applied the ultrasonic vibration.? Redraw the diagram indicating all the apparatus
  • Inline 162, the authors mentioned that the UVEF method increases the Tg, Why .? please explain.
  • Inline 165, what is Delta Vf ? explain
  • Give the Tg and Tx values for different as-cast and UVEF treated samples in the table.
  • Figure 5. The X-axis is different for each figure, make it uniform.
  • Figure 6. Split figure b into two and explain the y-axis legend in the description properly.
  • Line 207, it is difficult to understand what the authors want to convey 'fractures of the amorphous samples are not flat'?. usually, no fracture surface is flat.
  • Figure 8. Micron markers are not clearly visible, add clear micron marker in the images. Explain the inset in the figure description clearly.
  • Figure 10. Give the exact relaxation enthalpy values for the ac-cast and HEF treated samples.
  • Rewrite the author contribution and funding part
  • Reference 12 is not clear, add journal, volume, and page numbers.
  • Not enough citation and literature review is made in this article. Several researchers published good works in rejunuvation of BMgs, authors should cite them adequately.

Reviewer 2 Report

The article is about the rejuvenation of Zr52.5Cu17.9Ni14.6Al10Ti5 bulk metallic glasses (BMGs) by ultrasonic vibration-assisted elastic deformation. Authors have solved the problem of the BGMs brittleness due to a novel method of ultrasonic vibration-assisted elastic deformation.

However, there are some questions:

  1. I recommend authors to reduce captions for Fig 2 and Fig 9 e.g. To give decryption of abbreviation using one example.
  2. In the article, the abstract and the conclusion are almost the same things/ This is not entirely correct. The authors should rewrite a more detailed conclusion.

Reviewer 3 Report

The authors investigated the rejuvenation of Zr52.5Cu17.9Ni14.6Al10Ti5 bulk metallic glass via ultrasonic vibration-assisted elastic deformation. This document appears very interesting, although there are some comments that should be address.

1. There are some grammatical and spelling errors in the text. For example, BMG is spelled BGM on lines 26, 32, 34, and 39, 41, etc...

2. The authors should go into much more detail about the rejuvenation phenomenon in bulk metallic glasses. The references below should assist here:

Nature 2015, 524, 200-203, Journal of Nuclear Materials 2019, 526, 151771, Intermetallics 2018, 93, 141-147, Acta Materialia 2015, 86, 240-246.

3. For the experimental section, some more elaboration is needed on your methods. For example:

       a) What ASTM standard was used for the Vickers hardness? How many indents were performed? How far apart were in the indents space?

      b) What ion and energy was used to thin the TEM liftouts samples?

4. For Figure 2, please eliminate the noise from the X-ray diffraction patterns to ensure that there are no crystalline peaks in the data. 

5. Please provide a reference for the following discussion on lines 195-197 and 244-245:

"The decrease in microhardness is due to the increase in the atomic spacing of the amorphous alloy; the combined energy of the metal bonds is weakened and the amount of free volume increases"

" The ultrasonic vibration stress can create additional free volume and improve the
245 rejuvenation degree of amorphous alloys"

6. For Figure 8, please increase the font and graphic for the scale bar as it is extremely hard to read even when the screen size is increased.

7. For Figure 9(a), it appears that in some of the X-ray patterns, there may be some sharper peaks at lower angles. This result indicates that crystallization may have occurred during the hot-compressed elastic deformation. It is highly recommended that the authors remove some of the noise from the patterns to conclusively show that there are no crystalline peaks. 

8. Please provide the good of fit for the fitted curve from Figure 12.

Round 2

Reviewer 3 Report

The authors have sufficiently addressed my comments, therefore the paper is acceptable as is.